# The Role of Open Access Data in Geospatial Electrification Planning and the Achievement of SDG7. An OnSSET-Based Case Study for Malawi

**Alexandros Korkovelos** [1,*], **Babak Khavari** [1], **Andreas Sahlberg** [1], **Mark Howells** [1] and **Christopher Arderne** [2]

1   Division of Energy System Analysis, KTH Royal Institute of Technology, Brinellvägen 68, 10044 Stockholm, Sweden; khavari@kth.se (B.K.); asahl@kth.se (A.S.); mark.howells@energy.kth.se (M.H.)
2   The World Bank Group, Washington, DC 20433, USA; carderne@worldbank.org
*   Correspondence: alekor@kth.se; Tel.: +46-735-843-613

**Abstract:** Achieving universal access to electricity is a development challenge many countries are currently battling with. The advancement of information technology has, among others, vastly improved the availability of geographic data and information. That, in turn, has had a considerable impact on tracking progress as well as better informing decision making in the field of electrification. This paper provides an overview of open access geospatial data and GIS based electrification models aiming to support SDG7, while discussing their role in answering difficult policy questions. Upon those, an updated version of the Open Source Spatial Electrification Toolkit (OnSSET-2018) is introduced and tested against the case study of Malawi. At a cost of $1.83 billion the baseline scenario indicates that off-grid PV is the least cost electrification option for 67.4% Malawians, while grid extension can connect about 32.6% of population in 2030. Sensitivity analysis however, indicates that the electricity demand projection determines significantly both the least cost technology mix and the investment required, with the latter ranging between $1.65–7.78 billion.

**Keywords:** open data; electrification modelling; Malawi; OnSSET

---

## 1. Introduction

The 2030 Agenda for Sustainable Development has set the goal of universal access to electricity by 2030 (SDG7) [1]. The challenge is significant. It involves reaching populations with limited income, often living in sparsely populated areas, mostly in developing and least developed countries [2]. Selecting the optimal electrification approach is also difficult; grid vs. off-grid, fossil fuel vs. renewable, public vs. private investment are just a few examples. Coping with dilemmas of this nature—involving the deployment of big technological systems—requires thorough analysis of the social, technical, economic and political characteristics of the studied area or country [3]. This in turn, requires access to reliable data and information [4,5]; e.g., distribution and density of population settlements, electricity demand levels, resource availability, poverty rate and economic activity, distance from functional infrastructure (e.g., transmission and distribution network, roads, power stations) to name a few.

Despite progress, in most countries where universal electrification is still to be achieved, such official information is yet difficult to access [6]; these data are typically not covered by standard national energy statistics. The paucity of such information is one reason hampering electrification progress [7,8]. However, this situation is gradually being overcome with the increasing availability of new data and analytical tools, especially in the field of geospatial analysis. Geographic Information Systems (GIS) and remote sensing techniques are becoming openly available and can now provide a range of location-specific information that has not been previously accessible.

In the energy sector, the use of GIS data and associated analytical tools to conduct strategic planning remains at an early stage, yet such efforts have multiplied in recent years to further support both public and private stakeholders in prioritizing and rationalizing energy infrastructure investments [9]. From a public-sector perspective, GIS analytics are increasingly being used by governments and utilities to prioritize and sequence their grid extension efforts, as well as integrate off-grid solutions within national strategies aiming to achieve universal electricity access in a given timeframe (e.g., Tanzania [10], Afghanistan [11], Zambia [12], Madagascar [13]. From a private sector perspective, similar analytics are used to demonstrate the opportunity for supplying off-grid customers with decentralized energy services (market opportunity identification) and support subsequent operational roll outs (business models).

With this paper we aim to: (a) provide an overview of the main GIS data and modelling efforts aiming to support electrification planning and the achievement of SDG7; (b) discuss their role (especially if open) as providers of useful insights to difficult policy questions; (c) illustrate narrative through a case study of Malawi using an open-data-based and updated version of the Open Source Spatial Electrification Toolkit (OnSSET 2018), and (d) identify critical data/methodological gaps and suggest actions of future development.

## 2. GIS Based Electrification Planning

### 2.1. Open Access Data

The availability and quality of open access, publicly available GIS datasets has improved significantly over the past years; new datasets emerge conveying useful information regarding resource availability, status of infrastructure, social and economic characteristics of global populations. The following paragraphs present GIS datasets that have been (or can be) used in geospatial electrification analysis. A summarized list of useful GIS data for geospatial electrification modelling, providing status and gaps, is available in Appendix A.

#### 2.1.1. Energy Infrastructure

The development of effective—GIS based—electrification plans depends greatly on the availability of credible and up-to-date records of existing infrastructure in the area of interest. The distribution of grid network for example, is an important input parameter. To illustrate, unelectrified settlements might find it more economical to connect to the national grid if in close proximity to service transformers or medium voltage (MV) lines. In contrast, areas that are located far from grid network might find off-grid technologies (mini-grids or solar home systems) are a better alternative. Therefore, low quality (erroneous or inadequate) datasets of the grid network may have a considerable impact on the results of electrification models. Other infrastructure (and thus for planning their datasets) such as the road network, are equally important; take for example remote villages without access to proper roads. They might experience high logistic costs for certain technologies e.g., high diesel prices. Several efforts have been recorded over the past few years aiming at reducing infrastructure data gap; few of them are briefly described below.

A noteworthy initiative recording power plants worldwide is the Global Power Plant Database by World Resource Institute [14]; the dataset contains geo-located entries of 28,500 power plants from 164 countries, including information on capacity, generation, ownership, and fuel type. It is open and frequently updated. The Global Roads Open Access Data Set (gROADS), v1 [15] provides a range of road data from the 1980s to 2010. Unfortunately, most country data is not 'date stamped' and spatial accuracy varies. OpenStreetMap (OSM) comes to fill data gaps in several instances [16]; OSM is a big—and growing—repository of open geospatial data including various elements of infrastructure, including roads [17]. The World Bank, has developed an online data explorer that records existing and planned transmission and distribution lines over Sub-Saharan Africa and Middle East [18]. The explorer draws from a comprehensive dataset [19] including power lines ranging from sub-kV to

700 kV. It should be noted however that there is large variation in the completeness of data by country. The ECOWAS Centre for Renewable Energy and Energy Efficiency (ECREEE) has provided a similar dataset for West Africa [20].

These efforts collect, organize and redistribute existing data. However, for many countries datasets are incomplete and in some cases of uncertain quality; for example, metadata describing the content, its source and how it was derived is often incomplete or missing. In order to overcome selected barriers, new methodologies have been developed. The energy access team at Facebook has released a remote sensing base predictive model for more accurate MV network mapping; the model is open source with output—as of the time of writing—being available for six countries in Sub-Saharan Africa [21]. Note that [22] provides an adaptation of this work available as an executable, open source code. Other initiatives consider the use of machine learning techniques and artificial intelligence. For example, Development Seed has developed an open source pipeline to efficiently map the high-voltage (HV) grid at a country-wide scale. The method uses high resolution satellite maps (0.5 m/pixel), from DigitalGlobe Platform to identify HV-towers. Then machine learning algorithms are applied to predict the distribution of transmission lines between the towers. Results are available for Nigeria and Zambia [23]. Finally, other initiatives [24–26] have also been developed in this area; some are and some are not focused in Sub-Saharan Africa. As they are open however, they can be applied globally and provide the potential to overcome important data shortages.

### 2.1.2. Resource Mapping

Natural resource availability—such as sunlight for PV panels—is a significant decision parameter when choosing electrification options. Electrification solutions should take into account local conditions in order to be achieve long term sustainability [27]. Remote areas with abundant solar irradiance, far from oil supply, for example might be better served by photovoltaic systems rather than diesel generators. Similar logic is applied to other resources. So called 'big data' from Earth-orbiting satellites have enabled scientists to better assess resource availability on a global scale. This body of data, if processed properly can provide useful information for electrification projects as well. For example, a Global Horizontal Irradiation (GHI) dataset is available by [28]. They provide information regarding solar availability in a location (usually in $kWh/m^2/year$). Other datasets such as wind speed [29,30], Digital Elevation Models (DEM) [31], land cover [32–35], river network [36], drainage basins [37], water discharge flows [38,39] are also highly useful. Combination of those, can yield very useful outputs such as wind power density [29] or capacity factors [40], hydro potential maps [41], which in turn can provide insights for the development of successful electrification projects.

### 2.1.3. Socio-Economic

A critical challenge in current electrification efforts is to construct sustainable business models. That is, electrification projects (both private and public) need to be able to recover investment and operational costs and be profitable—or at least break even [42]. Information regarding the socio-economic context under which such projects are developed, is thus important during the design phase. The following paragraphs describe how GIS can help identify some of these characteristics and incorporate them into electrification modelling.

### 2.1.4. Population Density & Distribution

Population density and distribution maps are used to indicate where population resides, thus where there is potential residential demand [43]. The map type as well as spatial resolution determines the detail (and sometimes accuracy) of information. Gridded population datasets (similarly to any other raster layer) represent information in the form of grid cells. In this case, grid cell values indicate population headcounts or density in a specific time.

Worldpop [44,45] has developed gridded population layers for many Sub-Saharan African countries at 100 m spatial resolution; 1 km resolution layers are available at continental level. These

layers use interpolation techniques, which may give rise to inaccurate population estimates in certain cells; for example, some cells indicate population headcounts that have no physical meaning (e.g., less than 1). The Global Human Settlement layer (GHS) [46] suggests an alternative approach by indicating population values only in urban, peri-urban or rural areas; locations without population are eliminated. In similar manner, the Global Urban Footprint (GUF) [47–49] layer specifies in high spatial resolution (12 or 75 m) where settlements are located. The High Resolution Settlements Layer (HRSL) [50] provides population density maps in very high spatial resolution (30 m) but only for selected number of countries in Sub-Saharan Africa. Finally, [51] has developed a methodology that further processes the above datasets in order to provide more accurate vector type settlement layers for the case study of Tanzania.

### 2.1.5. Night-Time Lights

Night-time light (NTL) maps capture light sources on the surface of the Earth using satellite imagery. These can be a good proxy for assessing where electrified human settlements are, as they indicate light pollution. The Visible Infrared Imaging Radiometer Suite (VIIRS) dataset is available in raster format at 250 m spatial resolution; it provides the luminosity value in every cell; low value indicates that there is little visible light while higher values indicate high luminosity [52]. As of 2018, VIIRS provides annual composites for 2015 and 2016 as well as monthly composites for all years between 2012–2018. Its availability in monthly composites allows for detailed analysis of light sources and reduces the occurrence of false positives—areas that seem to be lit but in reality are not. Note that DMSP-OLS V4 [53] is VIIRS predecessor; it is available in raster format at 1 km spatial resolution and available for composites until 2012. It should be noted that DMSP-OLS V4 composites have been processed in order to provide stable light values over time series. Finally, Earth Observatory [54] also provides night light data at various spatial resolutions but without providing stable light composites.

### 2.1.6. GDP—Poverty Maps

Reference [55] developed a GIS layer presenting the Gross Domestic Product (GDP) in gridded format and on global scale for three intervals between 1990 and 2015 under 1 sq. km spatial resolution. The study uses primarily national GDP, PPP (purchasing-power-parity) values in constant 2011 international U.S dollars ($). In this instance, GDP illustrates the sum of gross value added by all resident producers in the economy plus any product taxes and minus any subsidies not included in the value of the products. It is calculated without making deductions for depreciation of fabricated assets or for depletion and degradation of natural resources. In addition, the study uses sub-national GDP, PPP values (for 82 countries) where available. The values were also converted in constant 2011 international U.S. dollars ($). These values were adjusted so that—when weighted by population—they total the GDP, PPP at the country level. By combining national and sub-national data, the global gridded GDP, PPP per capita maps were created. Poverty maps indicate the headcount ratio of population that lives below the poverty line (threshold usually being $1.25 or $2 per day) in an administrative area that can range from high level districts to lower level wards and municipalities. High resolution poverty maps (1 sq. km) have used geo-statistics in combination with GPS-located household survey data; such maps are however limited to a few countries. A combination of the aforementioned maps can provide very useful insights in electrification planning activities; they can be used as a proxy for economic activity or well-being in an area; or to create some sort of "heat map" indicating a better suited electricity access target per location.

### 2.1.7. Other

The list of open, energy related geospatial data is big and growing together with online GIS data platforms, map catalogues and repositories that make such datasets publicly available such as Energydata.info [56], OpenStreetMap [17], Google Earth Engine [57], IRENA global atlas [58], World Resource Institute [59], UN biodiversity lab [60], NREL GIS data & OpenEI [61,62], Earth Data [63].

Country specific GIS platforms have also been developed to support open data dissemination such as in Bolivia [64], Brazil [65], Kenya [66], Malawi [67], Uganda [68] and Namibia [69].

### 2.2. GIS Based Electrification Modelling Frameworks

The advent of geospatial information stimulated the development of modelling tools, methodologies and user interfaces that leverage on them so as to better support electrification planning decisions. The first tools that used GIS information in order to assess local resources and support techno-economic optimization included HOMER, RETScreen, SWERA UNEP, PVGIS, HOGA, DER-CAM [9]. Such tools are out of the scope of this paper as they mostly focus on the assessment of individual projects. The focus here is on what we shall term, the "second generation" of GIS modelling frameworks. The latter, utilize geospatial information and GIS software in order to support higher level electrification planning efforts. A short description of those most commonly used in electrification planning efforts is presented below. We shall also turn much of our attention to open access efforts, as they allow for reproducibility and thus can form the basis for scientific expansion.

IMPROVES-RE program (2007–2009) is one of the first efforts to support rural electrification activities with the use of a geospatial information. It is an open web-based platform aiming to support rural electrification projects and increase their impact on sustainable development and poverty alleviation in Burkina Faso [70]. It should be noted that IMPROVES-RE can be considered the predecessor of GEOSIM, a commercial tool that has been used for rural electrification planning in some countries (e.g., Tanzania [10]). GEOSIM is only marginally included in this review since it is not open source and relies on proprietary GIS software (Manifold).

Network Planner is a GIS based, open source [71] modelling framework for planning electricity infrastructure projects. Its underlying model identifies the optimal electrification technology mix for currently unserved demand centers; those include demand for households and other productive uses of electricity. Network planner uses a modified version of Kruskal's algorithm (minimum spanning tree) in order to find the maximum length of medium voltage lines for which grid extension is cheaper than the available off-grid options (solar home systems, diesel mini-grids) [72]; it does not however include biomass, wind and hydro as potential energy sources. Network Planner has been applied to Liberia [73], Ghana, [74] Kenya [75] and Senegal [76].

RE$^2$nAF is an open access web mapping application that enables geographically based exploratory analysis for off-grid electricity systems in the African continent. It overlays population settlements, infrastructure features (grid network, power plants and roads) and solar resources indicators (kWh/m$^2$) aiming to provide a comparison between diesel and PV based electricity costs for electrification [77]. All underlying GIS datasets have been made publicly available; results have been analysed and discussed in [27,78,79]. It shall be noted however that the underlying model is not available in the form of a customizable tool that could allow replication or modification by a broader user base, thus less capable of capturing specificities associated with individual projects.

The Reference Electrification Model (REM) [80,81] is an optimization tool designed to provide detailed engineering designs for electrification projects. It combines geospatial information with electricity demand and technology costs in order to estimate and compare different combinations of electrification modes (grid, mini-grids and stand-alone systems). Using satellite imagery, deep learning-based computer vision and clustering algorithms, REM can provide high level of granularity ranging from country to village level analysis. The model also offers the possibility to assess the impact of various factors such as demand levels, grid reliability, fuel and technology costs and cost of non-served-energy. REM has been used for electrification planning in India [82], regions of Rwanda and Uganda [83], Kenya and Colombia. REM (in liaison with its sibling tools GridForm and uLink [84]) offer a comprehensive modelling approach to rural electrification challenges. However, as in the time of writing the model is not yet open source.

IntiGIS is a plug-in application for ArcGIS that uses geospatial information in order to assess and compare the techno-economic performance of several electrification technologies; these include (a)

stand-alone systems (PV, wind, diesel), (b) mini-grid systems (diesel, hybrid—wind/PV/diesel) or (c) connection to grid MV lines. Results include numerical and cartographic values of each of the selected technologies, including the optimal levelized cost of electricity at each point of demand and sensitivity of various technical parameters. IntiGIS is distributed freely however its operation is dependent on proprietary software (ArcGIS). Results of its application are available for Ghana [85].

The Open Source Spatial Electrification Toolkit (OnSSET) [85] is a GIS based tool developed to identify the least-cost electrification option(s) between seven alternative configurations; grid connection/extension, mini grid systems (solar PV, wind turbines, diesel gensets, small scale hydropower) or stand-alone systems (solar PV, diesel gensets). OnSSET combines geospatial information related to infrastructure, resources, topology and socio-economic characteristics over a modelled area, in order to inform a tree search algorithm. The algorithm traverses iteratively through a sub-set of the tree nodes (un-electrified population settlements) using Locality-Sensitive Hashing (LHS) to identify the nearest neighbor and optimal electrification technology. Results indicate the optimal technology mix, capacity and investment requirements for achieving electricity access goals under pre-defined time series (This may include multiple time steps; minimum duration of a time step is one year). The model also considers a prioritization algorithm, which defines how electrification progresses over time. Findings can be presented in various GIS compatible formats such as interactive maps, graphs and tables. OnSSET has informed IEA's energy access outlook publications [2,86], UN estimates for all Latin American and African countries [87], as well as country studies for Ethiopia [88], Nigeria [89], Kenya [90], Afghanistan [11], Madagascar [13], Tanzania [91], Zambia [91] and Benin [92]. Electrification investment scenarios also feature for 56 countries in open access web-based platforms [87,91].

Other geospatial web-based applications are also available. The Off-grid Market Opportunities Tool uses geospatial information (such as population density, proximity to transmission and road network and others) to help private companies, governments, academia and civil society to develop a high-level view of where markets for off-grid electrification may exist to better inform decision-making [93]. The Nigeria Rural Electrification Plans (NESP) [94] web platform provides least-cost geo-spatial electrification plans for five Nigerian States including detailed standalone and mini-grid assessments together with grid extension modelling [95,96]. Myanmar off-grid analytics [97] is a web tool that maps village location in Myanmar and based on available GIS data (local resources and nearby infrastructure) provides information for potential investment in off-grid electrification technologies. Ghana Energy Access Toolkit (GhEA) [98] and ECOWREX GIS [99] are mapping tools used to monitor and evaluate renewable energy resources and energy access progress in the country using geospatial datasets.

Finally, there are few noteworthy methodologies that utilize geospatial information to inform electrification plans. They have not led to functional tools however they may be replicable. Tiba et al. [100] proposed—and applied in the case study of northeast Brazil—a GIS-based methodology that supports rural electrification. Kaijuka et al. [101] used GIS information to identify patterns of electricity demand in Uganda and suggest priority areas for energy investment in the country. Teske et al. [102] developed a comprehensive multi-sectoral approach aiming to provide universal access in Tanzania only though renewable energy based technologies. They used open access data and maps in order to visualize and analyze key parameters for the analysis of Tanzania's future energy situation. These included solar and wind resources, population density, access to electricity via the central power grid or mini grids, the distribution of wealth or the economic development projections as well as energy demand projection for each settlement.

*2.3. The Role of Open Access Data and Modelling Frameworks in Electrification Planning*

Naturally, data and tools are designed in different contexts and may serve specific purposes. Even though their capabilities and objectives may vary per case, we find that most electrification efforts follow a conceptual framework as illustrated in Figure 1.

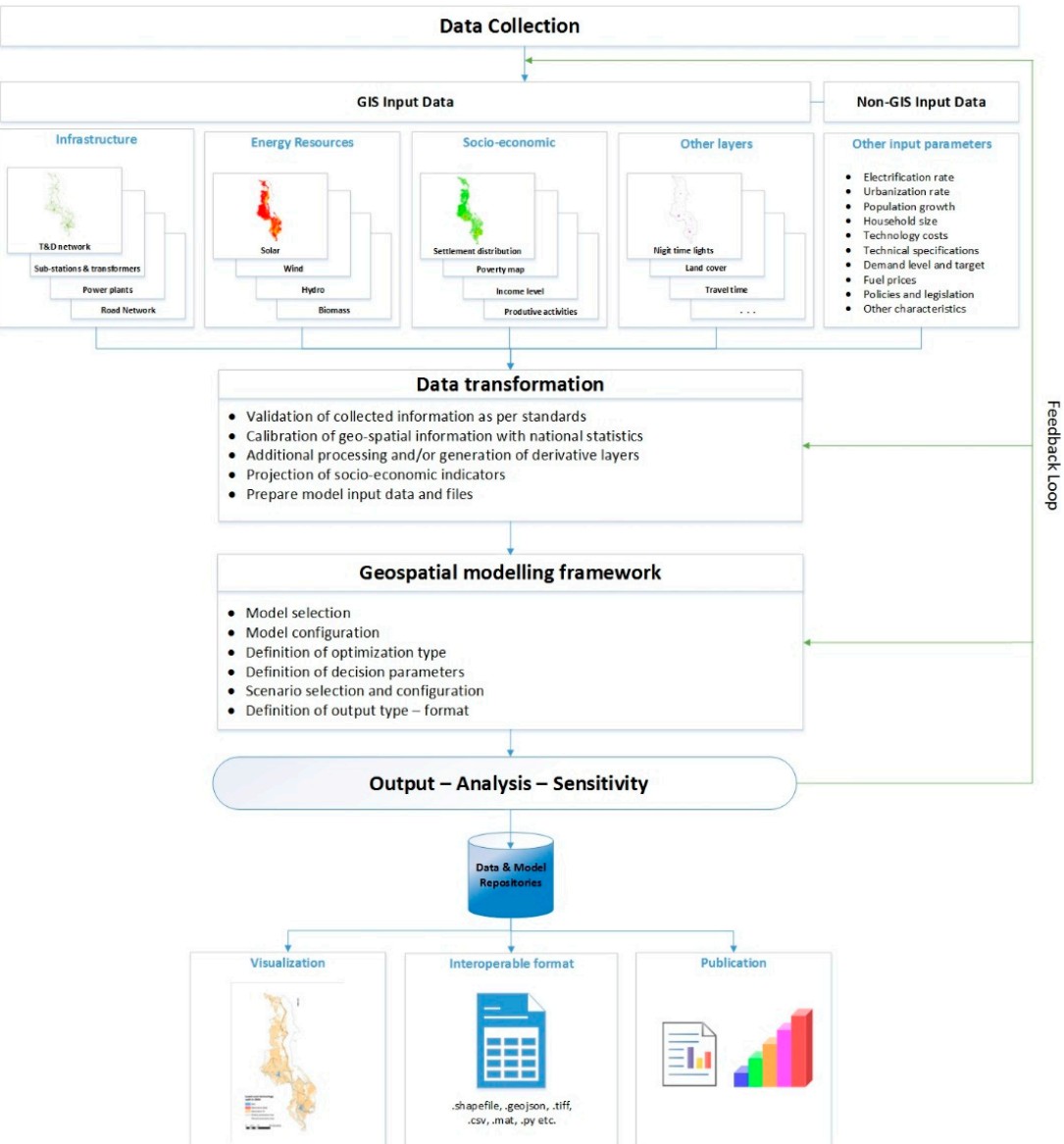

**Figure 1.** Conceptual flowchart of GIS -electrification modelling frameworks.

The flowchart in Figure 1 is far from exhaustive; it captures however the main components that we feel are crucial in GIS based electrification modelling. These include data collection, data transformation, model selection and configuration, result analysis and dissemination. Each component can be useful for policy making towards SDG 7; and here is where, we believe, open access can have the highest impact.

If open, such frameworks can enable the replicability and reproducibility of embedded processes as well as reusability of input/output data. In this way they can yield rapid techno-economic screening analyses—usually at low cost—in order to delineate the high level spatial contours of immediate (or intermediate) investment plans for electrification. They can also provide a test bed for cumbersome, long-term implementation roadmaps; support the decision making process; facilitate investment mobilization and speed up the implementation process. Finally, if transparently designed they can by audited by third parties; this is critical for assuring quality, control and demonstrating due diligence in administering public funds. With this in mind, we try to answer a set of questions commonly encountered in SDG 7 related planning and policy development activities. These may involve, among others, the following:

(1)     Where is the population located?

    a.     What is the population density and how are settlements distributed in the country?
    b.     What are the settlements' characteristics?

(2)     Which areas are currently electrified?

    a.     What is the level of access and use?
    b.     What is the expected/targeted electricity demand for different locations or types of settlements?

(3)     What is the optimal technology mix in order to achieve SDG7?

    a.     What equipment capacity is required?
    b.     What is the potential role of different types of electricity supply technology?

(4)     What is geospatial extent of the rollout electrification plan?

    a.     Where can the national grid reach?
    b.     Where do off-grid systems step up to provide access?
    c.     Which areas may get access to electricity first?

(5)     What is the cost of electrification?

    a.     What is the total investment required to achieve full access by 2030?
    b.     Where is investment most needed and in what form?
    c.     Where can households afford electricity and where should subsidization be considered?

The case of Malawi is selected since it is one of the countries with the lowest electrification rate in Sub-Saharan Africa. Following the conceptual flowchart presented in Figure 1 we set up an electrification investment scenario (EIS) using an updated version of the OnSSET modelling framework. We use entirely open access data, software and methods. It is to be noted that our findings are illustrative only. The aim is primarily to highlight the power of open access information and the positive impact they might have in supporting sustainable electrification policies.

## 3. Electrification Policy Insights for Malawi

### 3.1. Data Collection and Transformation

### 3.1.1. Question 1 on Population Distribution & Characteristics

Malawi is a south eastern African country with population of about 18.62 million people [103]. The population growth is 2.83%, leading to an estimated population of 26.03 million in 2030 [104] and the urbanization rate 4.41% [104] per year. The average estimated household size is 4.3 and 4.5 people for urban and rural settlements respectively [105]. Identifying the location and type of settlements is a very important first step in the geospatial electrification analysis, as it can help denote several other characteristics.

Population-based datasets exist mostly in the form of grids. A grid comprises a number of spatially identical cells. The size of the cell determines the spatial resolution or else the area it represents. Each cell is used to represent a settlement and usually comes along with an attribute that specifies either total number of people or population density. It is often the case that gridded population datasets use interpolation or extrapolation techniques in order to fill data gaps [106]. This can cause false positives/negatives—areas that seem to be populated but in reality are not or reverse—and skew the electrification results. In reality, human settlements have various geometries. In a perfect modelling world, human settlements would be spatially represented by delineated vector polygons (referred to

hereafter as population clusters) with full description of the settlement's characteristics (e.g., acreage, population, number and size of households). However, datasets of this nature are available only for limited locations. To overcome this, we introduce a new methodology aiming to delineated and attribute population clusters. This is achieved by using existing gridded population datasets and a set of open source geospatial processing tools. A step by step description of the methodology is presented in Appendix B. The methodology was tested upon the case study of Malawi. The derivative dataset yielded ~198,900 population clusters of various geometries and size as shown in Figure 2.

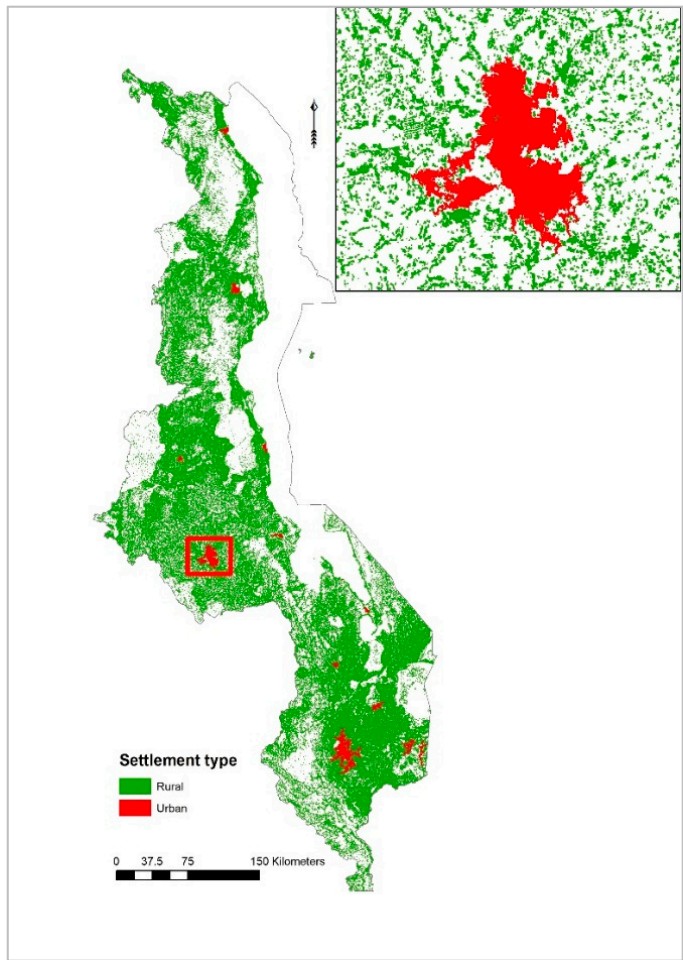

**Figure 2.** Characterization and spatial distribution of population clusters in Malawi as identified by the OnSSET model.

The aggregated population in the clusters was estimated as 17.19 million people, 7.65% lower than national statistics provide. The difference can likely be attributed to compounding uncertainty in geospatial processing and was mitigated through a calibration process. Once calibrated, each cluster was then characterized as either urban or rural based on information available at the GHS (S-MOD) layer. The layer provides a standardized distinction between: (a) urban centers, (b) urban clusters (peri-urban) and (c) rural settlements. For simplification, both (b) and (c) were considered as rural in this study. The process yielded 16 big urban clusters with an aggregated population of about 3 million people (in line with national statistics). The rest were identified as rural clusters.

Urban population settlements are often located closer to the existing grid network, they show higher population density, increased economic activity and (usually) higher electricity access rates and demand; the opposite applies to rural settlements [43]. In order to capture this dynamic, poverty and GDP data [55] were extracted to each cluster as shown in Figure 3a,b, accordingly. The poverty map

indicates the headcount poverty rate in each cluster; the GDP map indicates the estimated total gross domestic product in each cluster. Information regarding settlements' socio-economic characteristics can be an important indicator for the selection of an "appropriate" electrification technology that will assure long-term sustainability of this solution.

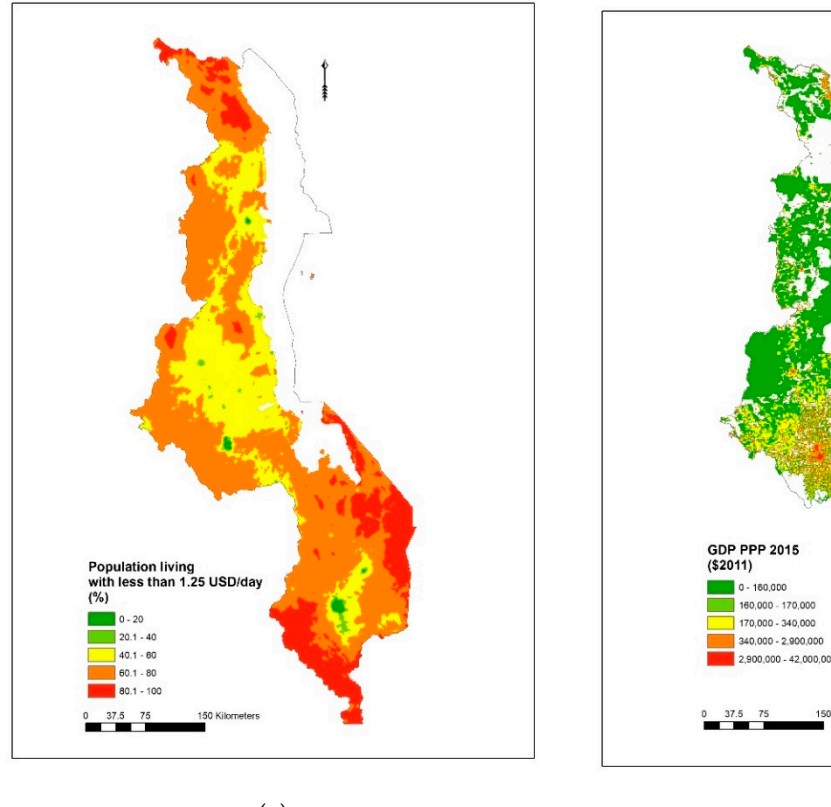
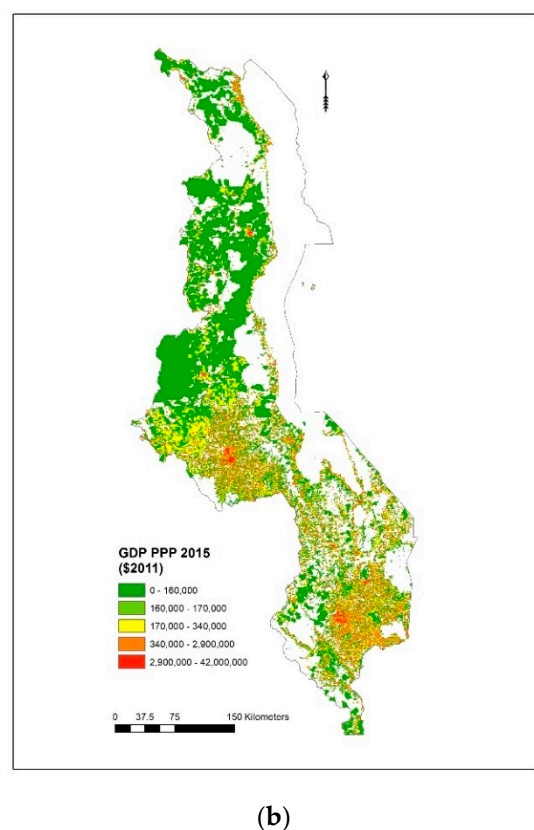

(**a**)                                                           (**b**)

**Figure 3.** Poverty rates (**a**) and estimated purchasing power parity Gross Domestic Product (GDP-PPP) in 2011 USD values (**b**) as distributed over population clusters in Malawi.

3.1.2. Question 2 on Current Electrification Status

The electrification rate in Malawi is among the lowest in the continent; it is estimated that about 49.2% of population living in urban areas has access to electricity while the rate is merely 3.2% in rural areas [107]. With the urban ratio in Malawi being roughly 17% [97], the national electrification rate stands at ~11%. Knowing where currently electrified clusters are located is an initial step needed for the electrification analysis. With OnSSET, already electrified clusters are used as anchor points for the electrification model. Once identified, they are, together with the known existing and planned grid lines, considered as starting points from which the grid network can be further extended. The location of electrified clusters and the access rate within those, is information often not easily accessed. Thus, in order to identify already electrified settlements rapidly a heuristic is added to OnSSET. That heuristic relies on a GIS-based multi-criteria evaluation. Note that this can easily be updated with actual figures when—and if—available (if can be the case, that with informal connections national statistics may be unhelpful in determining the extent of the electrification. National statistics may count only formal connections). The evaluation is based on five spatial attributes for each one of which a default threshold (the suggested values were reflective for Malawi; threshold values may vary per country) is defined as shown below:

(A)  Distance to service transformers (initial threshold, <1 km)
(B)  Distance to MV lines (initial threshold, <1 km)

(C)　Distance to HV lines (initial threshold, <5 km)

(D)　Nigh-time light intensity (initial threshold, >0)

(E)　Population (initial threshold, >300 people)

Priority factors can be assigned according to data availability and the level of confidence on the quality of the datasets. Independently, (A) serves as a priority proxy for identifying electrified locations. In case (A) is insufficient or not available, (B) is a considered a useful alternative. Finally, (C) might be used if none of the above is available. Yet, the use of (A), (B) or (C) alone might cause the selection of locations that are close to a line or transformer but not necessarily electrified; therefore, these layers shall be used in combination with (D) and/or (E). Note that in absence of both (A)–(C), the combination of solely (D) and (E) can yield alternative proxies. In fact, for the case of Malawi, 86.7% of all clusters with night-time light greater than zero are located within 1 km from a service transformer, and 96.7% are located within two km. Ideally as detailed surveys and measurement become available the validity of (and even the need for this) heuristic might be assessed.

While the authors had access to (A) and (B), these datasets were not openly available at the time of writing. For consistency with the narrative of this paper, we relied only on the use of (D) and (E) and identified 814 electrified population clusters. Then, for each one of these clusters, we calculated the ratio between lit and non-lit area (using NTL) and provided an estimate of the electrification rate within the cluster. Finally, we used an iterative routine in Python where the electrification rate in each cluster was calibrated so that the aggregated electrified urban and rural population matches the values indicated by national statistics. Results for Malawi are illustrated in Figure 4.

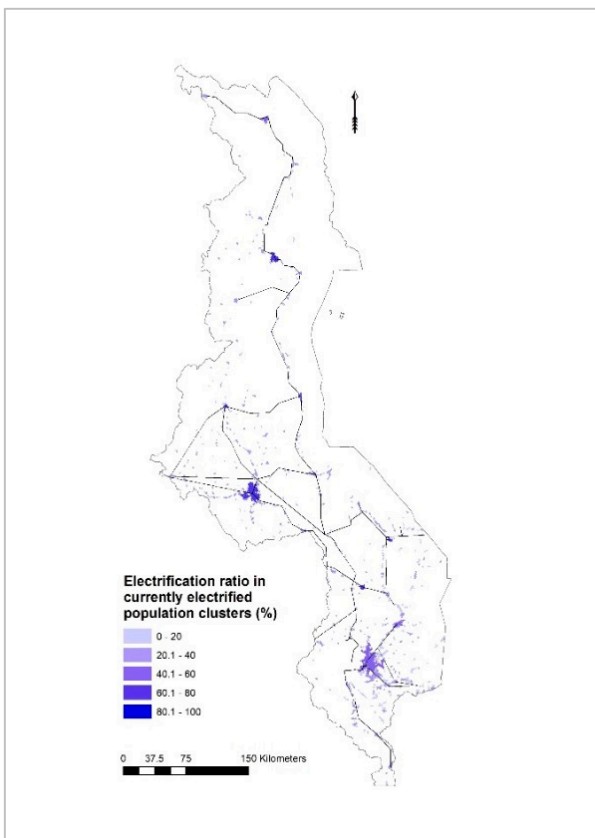

**Figure 4.** Distribution of settlements that indicate current access to electricity in Malawi. The multi-criteria evaluation yielded 16 urban and 798 rural electrified settlements with average electrification rates of 46.3% and 21.2% respectively.

According to the country's SE4ALL Action Agenda [108], the government in Malawi envisions that it will provide affordable and sustainable electricity services to all households at a level at least equivalent to Tier 1 (~38.7 kWh/household/year [109]) by 2030. Stimulated by this target and building upon the previous geospatial information, we prepare a map indicating targeted electricity levels per settlement as expected in Malawi by 2030. It should be noted that the current average household electricity consumption in Malawi is approximately 1072 kWh/year [108]. That is, all currently electrified settlements in Malawi were assigned a demand target equivalent to Tier 4 as in [109]. As illustrated in Figure 5, average electricity demand is expected to be higher in big urban clusters (Lilongwe, Blantyre, Zomba, Mzuzu); for the urban clusters identified in this analysis the median value of electricity demand was estimated at 32.4 GWh/year. For rural clusters the average electricity demand was estimated at 763 kWh/year.

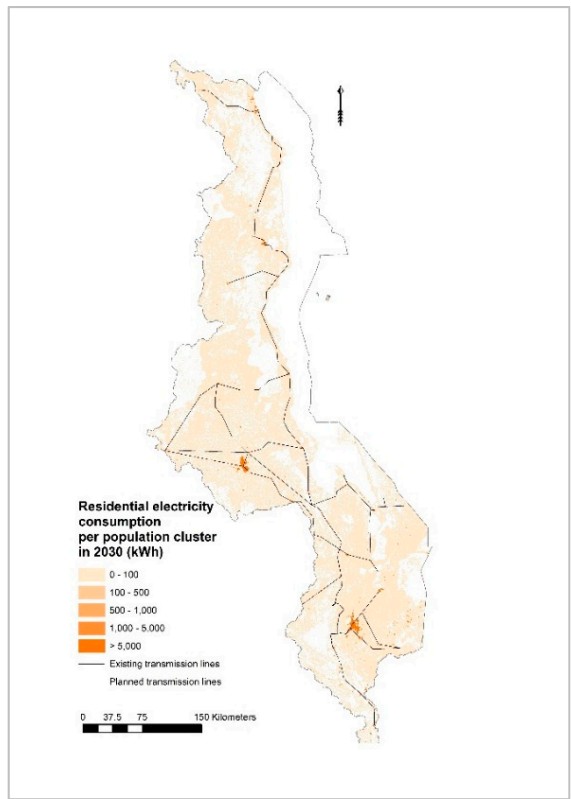

**Figure 5.** Distribution of the expected residential electricity demand per population cluster based on specified access targets (Tier 4 for urban and Tier 1 for rural clusters) in Malawi.

### 3.1.3. Additional Background Information

Malawi's current power system has a total installed generation capacity of about 361 MW with import capacity estimated less than 30 MW [110], whereas the country's current (actual and latent) demand is estimated to be as much as 700 MW leading to supply deficits [110]. According to the Master Plan and the rural electrification plan (MAREP), grid generation capacity will gradually increase to 1500 MW in 2020, 1859 MW in 2025 and 2519 MW in 2030 [108]. Grid extension currently plans to provide electricity to 31.6% of rural population by 2030 [108]. Beyond grid expansion, the government plans to electrify approximately 29.3% of rural population through solar home systems; and provide pico-solar systems to all the remainder (~39%) rural households by 2030. Other mini-grids are expected to electrify less than 0.1% of rural population [108]. Ramping up electricity access is a capital intensive process, especially in the rate under which this is expected to take place in Malawi. According to the SE4ALL Action Agenda, the cost of the suggested interventions for Malawi is estimated at $5.3 billion [108]. It shall be noted that similar electrification targets have been established in in

many developing countries nowadays [111]. Also, in a historical parallel, the electrification of 1.7 million farms in 1930s in the USA came at a cost of $321 million [112] (or ~$5.7 billion in 2018 values). The development of a cost effective and sustainable rollout plan for Malawi is therefore essential in order to avoid unnecessary sunk costs and sub-optimal investment portfolios.

## 3.2. Geospatial Modelling Framework Configuration

The electrification investment scenario was developed so as to reflect the background information presented in the previous section. Therefore, we assumed that urban settlements target achieving Tier 4 by 2030 while rural settlements aim at Tier 1. We assumed that all currently electrified settlements are grid-connected; in these settlements full access is achieved through grid intensification only. In contrast, the un-electrified settlements are assessed for electrification using all electrification technologies. The selection of electrification technology is based on the lowest cost required to meet the specified Tier in each cluster.

It was assumed that the electrification progress is gradual. That is, the national access rate was set to reach 50% in 2023 and 100% in 2030 [108]. This was achieved with the introduction of a time step function in OnSSET that allowed the definition of explicit access targets per time interval. In this case we selected two time intervals in the means of representing the first five-year investment perspective (2018–2023) and the overall target up to 2030. The time step function relies on a prioritization algorithm developed to first pick "Low hanging Fruit" sites. That is, the algorithm prioritizes grid intensification first; then it continues electrifying other settlements based on the lowest (to highest) investment cost per capita achieved (either grid or off-grid).

From a techno-economic standpoint the following assumptions were made. For the centralized grid generation, the average investment cost was assumed as 1874 $/kW based on the expected generation mix (this might include: large hydro at 1471.5 MW (58.4%), small hydro at 103.4 MW (4.1%), solar at 550 MW (21.8%), biomass (bagasse) at 46 MW (1.8%), coal at 300 MW (12%) and diesel at 48 MW (1.9%) [108]). in the country by 2030. Similarly, the grid generating cost of electricity was assumed as 0.076 $/kWh. It should be noted that this value does not reflect the customer tariff but the estimated cost of producing 1 kWh of electricity. (It is assumed that taxes and subsidies are applied ex-ante. Indeed, this needs to be the case in order to rationalise the level of subsidy required for electrification.) Other costs related to grid extension (T&D costs, losses and connection costs) were also considered. Techno-economic parameters for the off-grid electrification technologies included (a) investment cost ($/kW of installed capacity, including batteries), (b) operation and maintenance cost (% of investment cost per year), (c) capacity factor and (d) expected technology lifetime. Further, efficiency values and fuel costs were included for the diesel-based technologies. Finally, the discount ratio was set at 8%. A more detailed description of all assumptions is available in Appendix C. It should be noted that here for the purpose of this paper—introducing a cluster based approach to geospatial electrification—several parts of OnSSET were modified considerably. One of these, refers to the modelling of essential power components (e.g., type and size of substations, transformers, conductors). A more elaborate explanation of these modifications is available in Appendix D.

## 3.3. Output, Analysis and Sensitivity

The following section provides a brief analysis and visualization of key findings from the electrification investment scenario in relation to the policy questions posted in Section 2.3.

### 3.3.1. Question 3 on Optimal Technology Mix

The model suggests that national grid may electrify 32.6% of population in 2030. More specifically, grid extension may provide electricity to 6.3 million people by 2023 increasing to 8.5 million by 2030. It is also noteworthy that all new grid connections derive mainly from intensification, or else ramping up connections in already electrified locations. Extension of the grid network was only observed in a limited number of areas due to the low access target levels set in this scenario. In contrast, off-grid

technologies do play a very important role in this scenario. As indicated, the majority (67.4%) of the population in Malawi is expected to get access to electricity by off-grid stand-alone PV systems (Figure 6a).

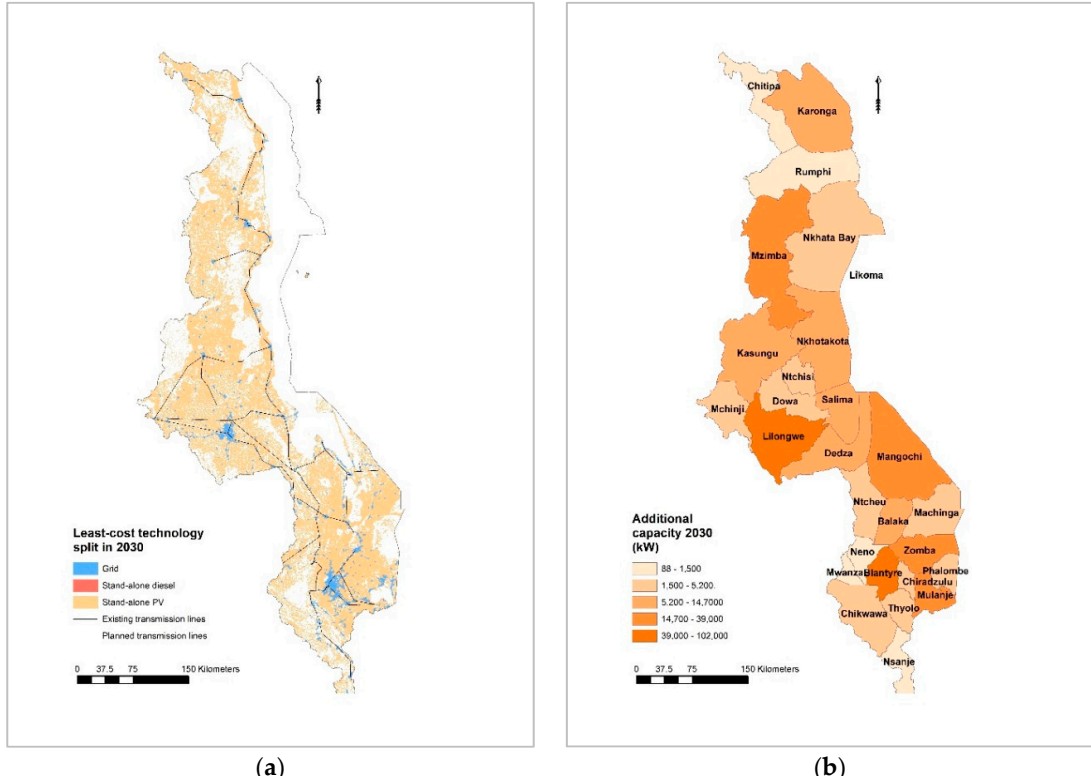

(**a**)                                                          (**b**)

**Figure 6.** Least cost technology split (**a**) and additional capacity (**b**) required per province to reach universal access to electricity in Malawi by 2030.

A few (<0.03%) stand-alone diesel systems were identified with no mini-grids (PV, wind, hydro or diesel) being included in the electrification mix in this scenario. In total, the country will need to increase the generating capacity by 351.8 MW by 2030 (168.1 by 2023 and 183.7 between 2023–2030) in order to meet the increased residential demand indicated by this scenario. From these, 23.9% shall derive from the deployment of stand-alone PV systems. That said and by assuming that grid generating capacity mix will be as described in Section 3.1, it is estimated that renewable technologies in Malawi can account for up to 89.5% of the additional generating capacity needed to achieve universal access goals by 2030.

### 3.3.2. Question 4 on Electrification Rollout Plan

From a geospatial perspective, national grid coverage is expected to cover 8817 km² or else about 14.4% of the populated land in Malawi. The majority of these areas are in close proximity to the existing network; in particular, 99.5% of the grid electrified population in 2030 (as per this scenario) is located within 5 km from the current grid network. Stand-alone PV systems have been identified as least cost electrification options in the rest of the country. In the districts of Nkhata Bay, Dedza, Ntcheu and Neno the electrified population by off-grid PV systems is expected to surpass 95%. Based on the time step function presented in Section 3.2 it was estimated that in the first five years of the analysis electricity service will reach about 8.76 newly electrified million people (Figure 7); from those about 51.5% will get access via off-grid systems while the rest through new grid connections (Figure 8). That is, grid connections will need to increase by a rate of 198,000 households per year until 2023 and slow down to 72,000 households per year between 2023–3030.

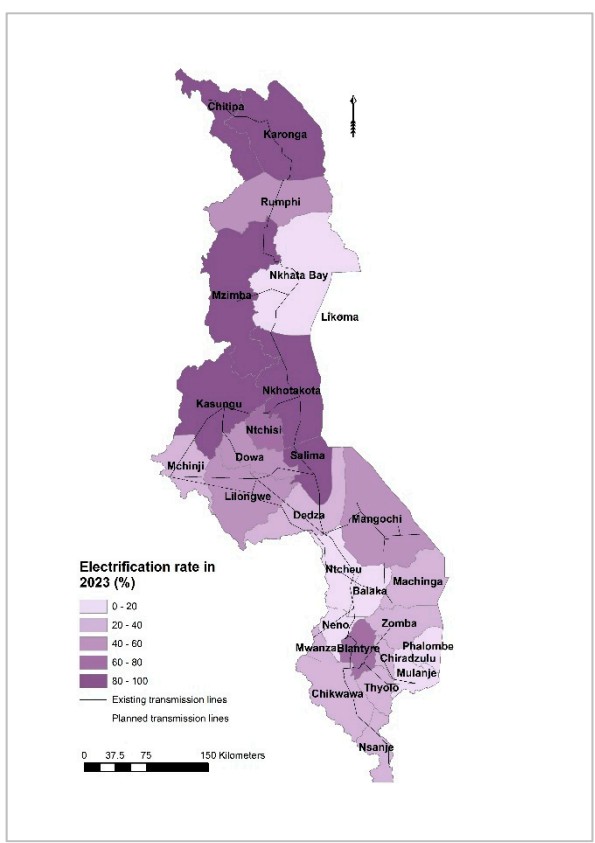

**Figure 7.** Percentage (%) of electrified population per province as in 2023.

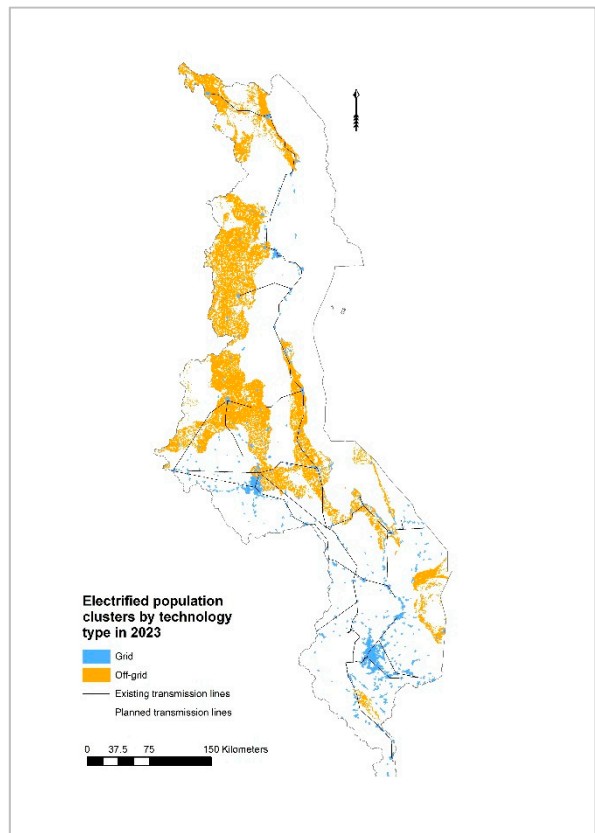

**Figure 8.** Grid vs. Off-grid split as per 2023 rollout plan estimated to electrify 50% of Malawians.

### 3.3.3. Question 5 on Cost of Electrification

The total investment required to achieve full electrification in Malawi by 2030, is $1.83 billion. New grid connections will require $1.48 billion. The investment cost per household varies depending on the distance to the transmission lines as well as the population in each settlement (Figure 9). The average cost of connecting to the grid amounted to $228.2 per person or else about $981 per household. It should be noted that these costs reflect mainly intensification of network; grid extension to new settlements even though slightly observed in this scenario might induce higher connection costs. Investment for decentralized technologies (stand-alone PV systems) is estimated to reach $351.9 million. The average connection cost for stand-alone PV systems was estimated at $26.3 per person or about $118 per household. Finally, for the few stand-alone diesel systems identified the average connection cost was estimated at $28.5 per person or about $128 per household. The distribution of required investment over Malawi is presented in Figure 10.

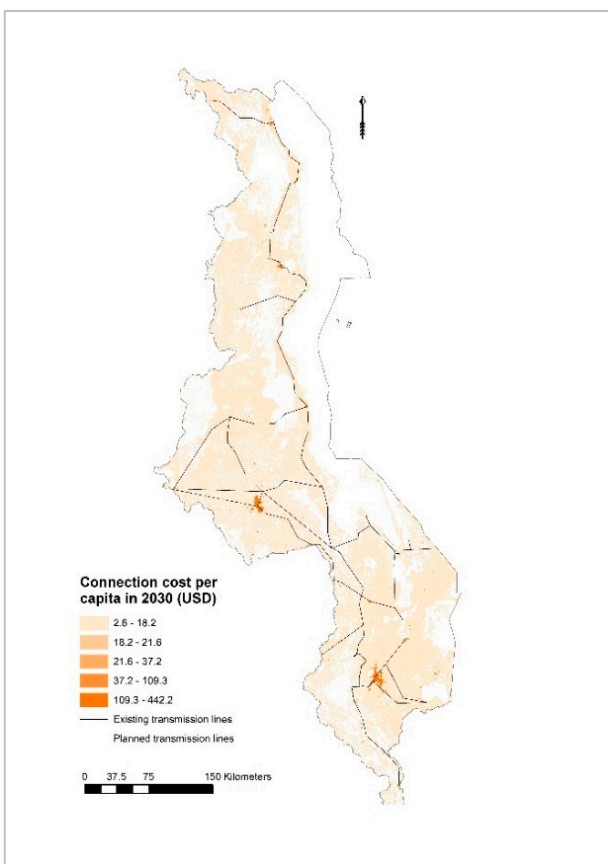

**Figure 9.** Connection cost per capita based on the least cost option identified in the selected electrification scenario.

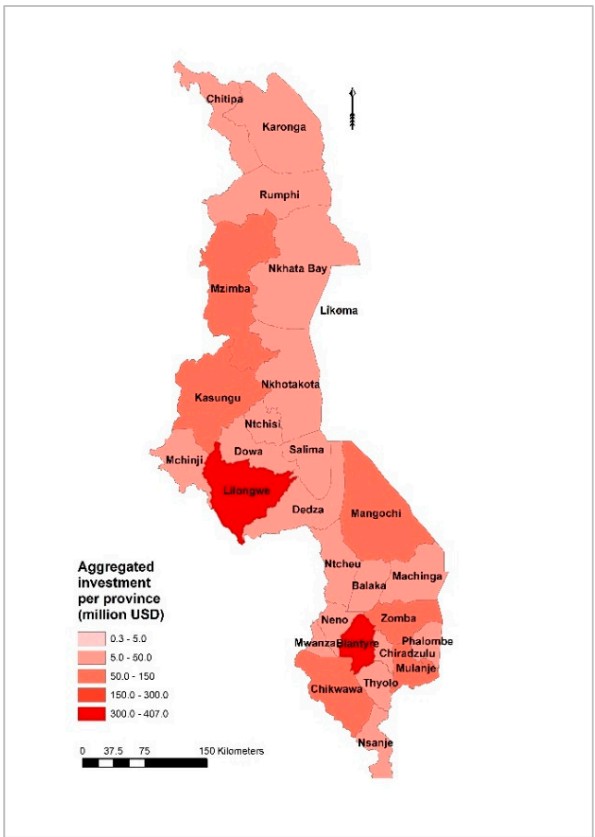

**Figure 10.** Investment requirements for the achievement of universal access as defined in the selected electrification scenario for Malawi by 2030. Results are aggregated per province.

3.3.4. Synthesis and Sensitivity Analysis

Comparing the results of this analysis with the government's estimates on achieving universal access in Malawi (Section 3.1), some noteworthy observations stand out. In both cases, grid connection is expected to provide electricity to approximately one third (31–33%) of the population. Also, the role of off-grid PV systems is crucial; solar home and pico-solar systems are expected to provide electricity to two thirds (67–69%) of the population by 2030. The additional capacity needed to achieved universal access based on the above statements is 267.5 MW for the grid and 84.2 MW for off-grid systems. However, government generation expansion plans will reflect general expansion vision that includes electricity demand not only for the residential sector. That also explains the disparity detected in terms of the total investment requirements. The electrification model estimated that $1.83 billion are needed to achieve the access target specified about a third of the government's estimates (~$5.3 billion). Lack of more detailed information on the rollout, investment plan from the government of Malawi limits the possibility of a more in depth comparison. Otherwise, based on these results the least cost electrification plan seem to be in alignment.

It is important at this point to highlight that any electrification analysis is subject to certain assumptions on the decision parameters. In this study we have selected to run a sensitivity analysis for six input parameters including population growth, electricity demand target, electrification rate in 2023, grid generation cost of electricity, PV cost and diesel cost. In total, ninety-six scenarios were generated and analysed indicating that the total investment requirements to achieve universal access to electricity in Malawi ranges between $1.65–7.78 billion. We find that the electricity demand target is the strongest determinant of both electrification investment and grid penetration in the total mix in comparison to the rest of parameters studied. A more detailed description of the findings is available in Appendix E.

## 4. Discussion

Open GIS data and modelling tools are increasingly being used in project development and planning in the energy sector. Their adoption and use can bring considerable advantages. It can provide a fast and cost effective way to map information that has a strong geospatial nature such as grid infrastructure, energy resources and settlement patterns. This can consequently, empower governments to effectively monitor progress, rationalize policy making and better inform strategic decisions in the energy field.

Electrification planning is no exception. The achievement of universal access to electricity is a crucial yet challenging task. It requires the motivation of significant financial resources in a timely and well-coordinated manner. This, given the rapid socio-economic changes and development particularly in currently unelectrified areas, makes the availability of good, up-to-date and consistent energy related information very important. This paper attempted to map existing data, tools and methods that have been commonly used to support SDG7 implementation efforts. It was observed that their number and importance has been progressively increasing over the past few years. Upon this, we provided key additions to the OnSSET methodology to form OnSSET 2018. Specifically, with this paper we have added an updated grid extension algorithm, a time step functionality and a new prioritization algorithm that allows the development of dynamic roll out plans for electrification. In addition, we introduce a restructured code basis that allows for a vector based approach of population settlements and the integration of new or upcoming geospatial datasets (MV lines, service transformers, poverty data, electricity demand for residential—e.g., Appendix F—as well as other productive activities). Despite that, limitations still exist and should be highlighted in the context of this analysis.

For geospatial data, the level of granularity is a key concern. Usually, open access data are available at low spatial or temporal resolution. Higher granularities are available either at a premium or under special agreement with the provider. Take for example T&D infrastructure; while at high level (e.g., HV lines) data have long been open and available for public consumption, at lower level (e.g., MV or LV lines) openly available data are scattered and inconsistent. This often leads to generalized assumptions, which in turn increase uncertainty in geospatial analysis. Reliability is another common concern. Open access geospatial data can be of unknown origin, questionable quality, poorly maintained, lack proper metadata or in some cases purposefully false. This makes quality assessment processes necessary for most practitioners before use, which can be time and resource consuming. Furthermore, despite progress, many socio-economic datasets potentially useful to electrification planning e.g., energy demand data, income level and distribution, energy expenditure, location of schools, health clinics and other productive nodes as well as mobile phone coverage are still limited or un-available in an open, geo-spatial format.

Similarly, the available GIS-based planning tools and methods, have one or more limitations: they are partially or fully proprietary; they focus only on rural areas and do not provide an overall electrification expansion indication for an entire country; they deploy a limited number of electrification technologies; they have restricted representation of demand; they lack a grid expansion algorithm or they do not account for a dynamic change of the bulk grid electricity supply. In OnSSET for example, the electricity demand is exogenous (layers imported from external calculations) and provide only an educated estimate. In addition, demand currently reflects only residential electrification targets. The model considers a set of static end-states (myopic optimization) thus, it does not use perfect foresight. Load profile is also represented on the basis of peak-to-average demand; that is reliability is incorporated but not optimized for. Despite its limitations, the basic OnSSET model is simple and open, allowing for a more tailored analysis to suite needs as needed—including improving all the above. Looking forward, we identify a clear need for synergies between the existing initiatives in the geospatial electrification field. The development of a single tool that incorporates all dimensions mentioned above could be theoretically feasible. Yet, we suggest that the development of a collaborative, open-source environment including interoperable data and tools with different characteristics might be more desirable. It could conceivably cover a wider range of applications and solutions—as well as harness

a greater volume of analysts and communities. This consequently would require that both data and tools should be democratized so that electrification analytics become accessible to more actors. Thus, global partnerships that promote collaboration between stakeholders who collect, create, manage or use geospatial data are particularly needed. A notable effort in this regard is the multi-country, multi-agency 'round-table' effort championed by DFID, as well as the fledgling Open Tools, Integrated Modelling and Upskilling for Sustainable-development (OpTIMUS) community of practice. These, might be enhanced through an inclusive, open and scalable platform that allows universal access global data layers as well as customizable modelling solutions. Such a platform would help build spatial literacy in the field of energy access and enable better decision making to deliver SDG7.

## 5. Conclusions and Final Remarks

As elements in a growing energy planning ecosystem, open access geospatial data and models have started a paradigm shift; a shift that constitutes a significant improvement over conventional planning efforts. Their availability and accessibility can help policy makers, government agencies, investors and project developers to overcome paucity of information and better inform decision making mechanisms. Despite its limitations, we hope that this study will help setting up new ground in the field of geospatial electrification planning and accelerate progress against the achievement of SDG7. Thus, the code basis of the updated electrification toolkit (OnSSET 2018) as well as input/output files for all electrification scenarios included in this paper are publicly available in [85] and open to review, update and/or reproduction.

**Author Contributions:** Conceptualization, A.K. and M.H.; Methodology, A.K., A.S. and B.K.; Software, A.K., B.K., A.S. and C.A.; Validation, M.H.; Formal Analysis, A.K.; Investigation, A.K.; Resources, A.K. and B.K.; Data Curation, A.K., B.K. and A.S.; Writing—Original Draft Preparation, A.K.; Writing—Review & Editing, A.K., M.H., C.A.; Visualization, A.K. and B.K.; Supervision, M.H.; Project Administration, M.H.; Funding Acquisition, M.H. and A.K.

**Funding:** This research was funded by the World Bank under the contract number 7185716.

**Conflicts of Interest:** The authors declare no conflict of interest.

## Appendix A. Listing and Gaps of GIS Data in Geospatial Electrification Modelling

**Table A1.** GIS data gap analysis in electrification modelling.

| # | Dataset | Type | Description | Status |
|---|---------|------|-------------|--------|
| | | | **Infrastructure** | |
| 1 | High Voltage (HV) lines | Line vector | Spatial distribution of (Existing & Planned) the transmission network. HV capacity definition depends on the country but usually refers to lines above 69 kV. | Publicly Available |
| 2 | Medium Voltage (MV) lines | Line vector | Spatial distribution of the medium voltage transmission network. What is defined as medium voltage depends on the country but usually refers to lines between 11–69 kV. | Not publicly available |
| 3 | Substations | Point vector | The location of currently available substations. Capacity and type should be provided as attributes. | Publicly Available |
| 4 | Transformers (primary or service) | Point vector | The location of currently available transformers. Capacity and type should be provided as attributes. | Not publicly available |
| 5 | Road Network | Line vector | Existing & planned road infrastructure. The road network may include major roads such as highways, primary and secondary roads. Detail should go as low on the road scale as can accommodate a pickup/truck. | Publicly Available |
| 6 | Power Plants (Existing & Planned) | Point vector | The locations of existing and planned power plants. It is important that the dataset includes attributes regarding each plant's minimum capacity. | Publicly Available |

**Table A1.** *Cont.*

| # | Dataset | Type | Description | Status |
|---|---------|------|-------------|--------|
| | | | **Energy Resources** | |
| 7 | Global Horizontal Irradiation (GHI) | Raster | Provide information about the Global Horizontal Irradiation (kWh/m$^2$/year) over an area. | Publicly Available |
| 8 | Small scale Hydropower potential | Point vector | Points showing potential mini/small hydropower potential. The layer shall include information regarding the location of potential sites, power output (kW), head (m) and the discharge (m$^3$/year). | Publicly Available |
| 9 | Wind speed or Power Density | Raster | Provide information about the wind velocity (m/sec) over an area. This layer may be substituted by wind power density maps (W/m$^2$). | Publicly Available |
| 10 | Biomass | Raster | Current and potentially productive agricultural activity as an indicator of agricultural residues. | Publicly Available |
| | | | **Socio-economic** | |
| 11 | Population density and distribution | Raster or vector | Spatial quantification of the population for a selected area of interest (usually country or continent). | Publicly Available |
| 12 | Administrative Boundaries | Polygon vector | Includes information (e.g., name) of the country(s) to be modelled and delineates the boundaries of the analysis. | Publicly Available |
| 13 | Residential demand | Raster | Layer that indicates electricity demand for residential sector | Not publicly available |
| 14 | Poverty maps | Raster or vector | Poverty maps stating the headcount rate (%) for the population below the poverty line. The poverty line used should be clearly stated. | Publicly Available (to some extent) |
| 15 | Income level or expenditure indicators | Vector or Raster | The income level or energy expenditure in an area ($/km$^2$). Map can be either in raster format or vector data on the basis of administrative areas. | Not publicly available |
| 16 | Gross Domestic product (GDP) | Raster | GDP map showing the purchasing power parity over an area. Map can be either in raster format or vector data on the basis of administrative areas. | Publicly Available |
| 17 | Human Development Index (HDI) | Raster | Providing information regarding the Human Development Index in an area of interest. Map can be either in raster format or vector data on the basis of administrative areas. | Publicly Available |
| 18 | Productive uses—Education facilities | Point vector or raster | Locations of schools as vector with relevant attributes (e.g., size of school, no of students, electricity needs/consumption). | Not publicly available |
| 19 | Productive uses—Health facilities | Point vector or raster | Locations of health clinics/hospitals as vector with relevant attributes (e.g., type or size of clinic, electricity needs/consumption) | Not publicly available |
| 20 | Productive uses—Commercial | Point vector or raster | Locations of commercial units (mines, businesses et.) as vector with relevant attributes (e.g., type or size, electricity needs/consumption.). | Not publicly available |
| 21 | Productive uses—Agricultural demand | Raster | Electricity demand layer (e.g., raster) indicating per capita (kWh/pp/year) or per settlement values (kWh/settlement/year) and is related to agriculture (e.g., pumping irrigation, post-harvesting). | Not publicly available |
| | | | **Other** | |
| 22 | Travel time | Raster | Visualizes spatially the travel time required to reach from any individual cell to the closest urban centre. The unit shall be in minutes/hours. | Publicly Available |
| 23 | Elevation | Raster | Filled Digital Elevation Model (DEM) maps. | Publicly Available |
| 24 | Land cover | Raster | Land cover classification. Currently OnSSET uses 17 classes as described in [32]. | Publicly Available |
| 25 | Slope | Raster | A sub product of DEM. The slope map visualizes the terrain slope in degrees. Any slope map that is to be used has to provide the slope in degrees. | Publicly Available |
| 26 | Night-time Lights | Raster | Night-time light maps showing light pollution. The map has a relative scale for the intensity of light. | Publicly Available |

## Appendix B. Methodology to Generate Population Clusters Using the High Resolution Settlement Layer and GIS Processing

The following methodology (Figure A1) has been developed in order to create population clusters based on open access population datasets from the HRSL and a series of processes developed in QGIS, an open source desktop geographic information system application.

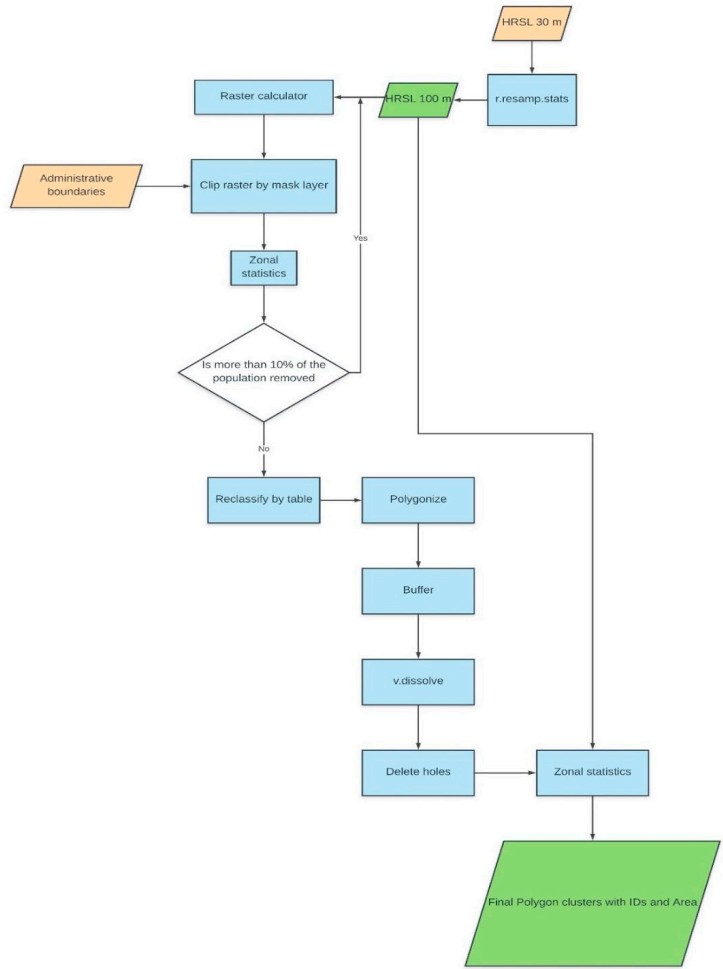

**Figure A1.** Methodological flowchart of creating population clusters using the High Resolution Settlements Layer and Geographic Information Systems processing.

### Appendix B.1. Resampling Population Layer

The original spatial resolution of HRSL is 900 m$^2$. In the case of Malawi this translates to 3.2 million grid cells that have to be processed in the GIS environment. This is problematic due to (a) computational limitations of the GIS software used (QGIS) and (b) memory/running time complications of the electrification model used (OnSSET). Therefore, reducing the spatial resolution (resampling) of HRSL, is a sensible—and highly suggested—first step in the process. A final resolution of 0.1 km$^2$/10,000 m$^2$ is a good compromise as it will significantly reduce computational limitations while maintaining a good level of granularity. Lower resolution than 0.1 km$^2$ (or 10,000 m$^2$) will cause undesirable distortion of the layer's values and therefore is not considered as a viable option.

*Suggested tool in QGIS: "r.resamp.stats".*

Notes/Comments: This tool is part of GRASS GIS and enables the user to resample raster datasets. As of the time of writing, this is the only tool included in QGIS 3.2 that allows for increasing the cell size while automatically aggregating the raster values.

*Appendix B.2. Removing Redundant Cells*

HRSL population density values derive from interpolating recent census data [113]. This creates grid cell "neighborhoods" in the raster that have the exact same value to the 16th digit. These grid cells are considered false positives and thus shall be removed. In order to eliminate falsely populated grid cells, a threshold value is defined through an iterative process described below.

Step 1. Calculate the total population in the area of interest.

*Suggested tool in QGIS: "Zonal statistics" from the QGIS package.*

Step 2. Initialize the threshold value; the initial value can be anything within the density range in the area of interest.

Notes/Comments: The threshold value can be determined by examining the distribution of pixel values for the raster dataset. Also, removing low populated grid cells increases the share of coinciding built-up areas in comparison to Google map tiles.

Step 3. Zero out all grid cells with raster value below the threshold.

*Suggested tool in QGIS: "Raster calculator" e.g., (HRSL > 6) * HRSL removes all values below 6.*

Notes/Comments: The "Raster calculator" rounds the coordinates for the raster to the first six digits. Therefore, there might be a slight offset between the datasets after using the tool. Since this raster is the base of the clusters the raster calculator should be used twice; once to multiply by one and once to carry out the operation described above. This way there will not be any offset between the different datasets used in the analysis.

Step 4. Re-calculate the total population in the area of interest. If this loss is larger than 10% repeat again from Step 2 using a lower threshold value. Repeat until loss is acceptable.

*Appendix B.3. Reclassify HRSL*

The re-classification of the HRSL is necessary for the population clusters to be formed uniformly during the next step. This process creates the conditions for all adjacent cells to become part of the same cluster (Figure A2-left). If not re-classified, the clusters will be comprised by multi-part polygons as shown in Figure A2-right.

*Suggested tool in QGIS: "Reclassify by table" from the QGIS package.*

Notes/Comments: There is a number of tools that can be used in order to reclassify a raster layer in QGIS. This specific tool is from the same package as the raster calculator and therefore it does not create any further distortion or offset.

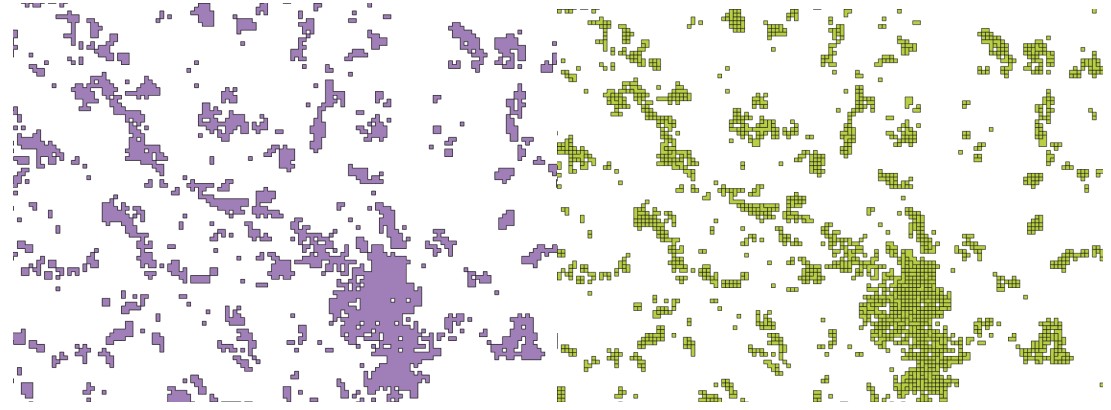

**Figure A2.** Uniform (**left**) and multi-part (**right**) population clusters created from HRSL.

*Appendix B.4. Convert the HRSL Raster to Vector Polygons*

In this process QGIS is used in order to convert the format of the processed HRSL from raster to vector polygons.

*Suggested tool in QGIS: "Polygonize" from the GDAL package.*

*Appendix B.5. Buffering Polygons*

A buffer of 10 m is applied to the polygons. This is due to QGIS treating polygons with one common corner as separate even in cases in which they touch. By applying a small buffer, it is ensured that these polygons are overlapping.

*Suggested tool in QGIS: "Buffering vectors" from the GDAL package.*

*Appendix B.6. Dissolving Polygons*

Dissolving the polygons ensures that overlapping polygons from the previous step are all merged.

*Suggested tool in QGIS: "v.dissolve" from the GRASS package.*

*Appendix B.7. Remove Gaps and/or Slivers inside Polygons*

Converting a raster layer to vector polygons as in previous step, can generate gaps and slivers to some of the polygons due to holes in the raster layer and due to the buffering process. These need to be removed/dissolved so that uniform population clusters are created.

*Suggested tool in QGIS: "Delete holes" from the QGIS package.*

Notes/Comments: It is important to only cover holes and slivers caused by the clustering process and not holes naturally occurring holes (e.g., lakes, forests etc.). Therefore, a maximum area is specified in the tool and all holes smaller are deleted.

*Appendix B.8. Assigning Population Values to Clusters*

Due to the population being reclassified when generating the clusters there is no population value connected to the clusters. In order to assign population values the raster values are aggregated for every cluster.

*Suggested tool in QGIS: "Zonal statistics" from the QGIS package.*

**Appendix C. Techno-Economic Input Parameters in OnSSET**

**Table A2.** Techno-economic parameters for off-grid technologies included in the electrification analysis.

| Plant Type | Indicative Capacity (kW) | Investment Cost ($/kW) | O&M Costs (% of Inv. Cost/Year) | Efficiency | Capacity Factor * | Life (Years) |
|---|---|---|---|---|---|---|
| Mini-grid diesel | 100 | 721 | 10% | 33% | 0.7 | 15 |
| Mini-grid hydro | 1000 | 5000 | 2% | - | 0.5 | 30 |
| Mini-grid PV | 100 | 4300 | 2% | - | Obtained by model | 20 |
| Mini-grid wind | 100 | 2500 | 2% | - | Obtained by model | 20 |
| Stand-alone diesel | 1 | 938 | 10% | 28% | 0.5 | 10 |
| Stand-alone PV | 0.3 | 5,500 | 2% | - | Obtained by model | 15 |
| Diesel pump price | 1.2 ** | $/litre | | | | |
| Connection cost Mini-grid | 125 | $/household | | | | |
| Connection cost Stand-alone | 0 | $/household | | | | |
| Discount rate | 8 | % | | | | |

* An indicative capacity factor was specified externally for diesel based technologies and hydro; capacity factor values for solar and wind were estimated by the model based on natural resource availability at each location; ** The diesel pump price was assumed at ~1.2 $/liter (900 MWK) [114,115]; exchange ratio used as $1 to 714.3 MWK.

**Table A3.** Techno-economic parameters related to the operation of the centralized grid and its extension process.

| Parameter | Value * | Unit |
|---|---|---|
| HV cost (69 kV) | 28,000 | $/km |
| MV cost (33 kV) | 13,000 | $/km |
| MV amperage limit | 8 | Ampere |
| LV cost (0.2 kV) | 10,000 | $/km |
| Max LV line length | 0.5 | km |
| Load moment | 9643 [116] | For 50 mm aluminum conductor under 5% voltage drop (kW m) |
| Service transformer (50 kVA) | 3500 | $ |
| Max nodes per transformer | 300 | nodes |
| MV to MV substation (400 kVA) | 10,000 | $ |
| HV to MV substation (1000 kVA) | 25,000 | $ |
| MV max reach | 50 | km |
| Base to peak ratio | 0.5 | - |
| Connection cost per household | 150 | $ |
| T&D losses | 10% [117] | of capital cost/year |
| O&M costs of distribution | 2% [117] | of capital cost/year |
| Grid extension cost ratio | 10 | % |
| Power factor | 0.9 | - |
| System life | 30 | years |
| Discount rate | 8 | % |

* The cost of grid components adopted in this study was primarily based on reference values provided by [118,119]; the selected values reflect authors' best estimate for the case of Malawi and they are only indicative.

**Table A4.** Expected generation mix, investment costs and generating costs for the expected centralized grid technologies in Malawi in 2030.

| Technology Type | Expected Capacity (MW) in 2030 [110] | Share (%) | Investment Cost * ($/kWe) | Generating Cost ** ($/kWh) |
|---|---|---|---|---|
| Hydro (large) | 1471.5 | 58.4% | 1929 | 0.05 |
| Hydro (medium/small) | 103.4 | 4.1% | 5025 | 0.08 |
| Solar (utility) | 550 | 21.8% | 935 | 0.15 |
| Coal | 300 | 11.9% | 2080 | 0.08 |
| Diesel | 48 | 1.9% | 708 | 0.23 |
| Biomass | 46 | 1.8% | 4105 | 0.07 |
| Average weighted | 2518.9 | 100% | 1874 | 0.076 |

* Estimated overnight capital costs were retrieved from [117,120]; ** Estimates for Hydro, Solar and Biomass were retrieved from [121]; estimates for coal from [120] and for diesel from [122].

## Appendix D. Updated Grid Extension Algorithm

The following paragraphs describe the modifications induced on the grid extension algorithm in OnSSET 2018. As of the previous version of the tool, the grid extension algorithm was based on the square geometry of a grid mesh with equal sized grid cells being adjacent to each other [123]. The integration of population clusters in the analysis, required the modification of the algorithm so that is it able to process vector data (polygons) of various geometry, size and spatial orientation. We describe the updated process in five distinctive steps.

Step 1. Sizing transmission lines (HV or MV)

As a first step, the algorithm decides the type of extension line (HV or MV) to be used to connect a settlement; the decision is based on two parameters as presented in (A1):

$$transmission\_line\_type = \begin{cases} \text{MV,} & grid\_distance \leq \text{MV}_{max\_reach} \, || \, peak\_load \leq \text{max\_MV\_load} \\ \text{HV,} & \text{otherwise} \end{cases} \quad \text{(A1)}$$

where:

$$peak\_load = \frac{\frac{Cluster\_electricity\_demand \div (1-T\&D \ losses)}{8760}}{Base \ to \ peak \ load \ ratio} \qquad (A2)$$

$$max\_MV\_load = MV_{type} \times MV_{amp\_limit} \times \frac{HVcost}{MVcost} \qquad (A3)$$

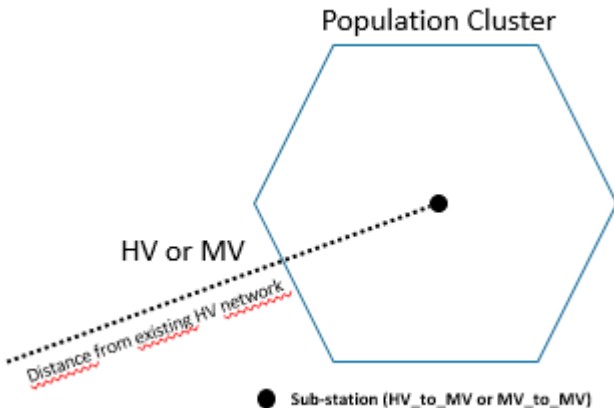

**Figure A3.** Estimating transmission line length from existing grid network.

Then, the mileage of additional transmission lines required to reach the cluster is estimated using (A4)–(A7) as follows:

$$transmission\_line_{km} = grid\_distance \times No\_of\_transmission\_lines \qquad (A4)$$

where:

$$No\_of\_transmission\_lines = \frac{peak\_load}{line_{amperage} \times line\_type} \qquad (A5)$$

$$line_{amperage} = \frac{substation\_type}{transmission\_line\_type} \qquad (A6)$$

$$substation\_type = \begin{cases} MV \ to \ MV, \ line\_type : MV \\ HV \ to \ MV, \ line\_type : HV \end{cases} \qquad (A7)$$

Step 2. Sizing transformers and connection to sub-station

Then, the algorithm estimates the number of service transformers required to provide full coverage of the population cluster:

$$No\_of\_service\_transformers$$
$$= max \left\{ \frac{S_{max}}{service\_transformer\_type}, \right.$$
$$\left. \frac{total\_nodes}{nodes\_per\_transformer_{max}}, \frac{cluster's \ area}{transformer\_area\_coverage_{max}} \right\} \qquad (A8)$$

where:

$$S_{max} = \frac{peak\_load}{power\_factor} \qquad (A9)$$

$$transformer\_area\_coverage_{max} = \pi \times LV\_line\_length_{max}^{2} \qquad (A10)$$

$$total\_nodes = \frac{cluster\_population}{No\_of\_people\_per\_household} + productive \ nodes \qquad (A11)$$

$$No\_of\_people\_per\_household = \begin{cases} 4.5, \ cluster \ type : Urban \\ 4.3, \ cluster \ type : Rural \end{cases} \qquad (A12)$$

The transformer load is the sum of the load of all households connected to a single transformer:

$$transformer\ load = \frac{peak\_load}{No\_of\_service\_transformers} \tag{A13}$$

It should be noted that the transformers are assumed to be evenly spaced within a cluster, thus the average distance from the service transformer to the substation is 2/3 of the cluster's radius, and the average distance between two service transformers is twice the transformer radius:

$$transformer\_distance_{average} = \frac{2}{3} \times cluster\_radius \tag{A14}$$

$$cluster\_radius = \sqrt{\frac{cluster\_area}{\pi}} \tag{A15}$$

$$transformer\_radius = \sqrt{\frac{\frac{cluster\_area}{No\_of\_service\_transformers}}{\pi}} \tag{A16}$$

If the estimated load moment is larger than 9643 (see Appendix C) an MV line is used to connect the service transformer to the substation; if not, a LV line is used. If connected by LV lines, each service transformer is assumed to have its own connection to the substation. With MV lines, multiple transformers may be connected in series:

$$load\_moment = transformer\_distance_{average} \times transformer\_load \tag{A17}$$

$$connection\_line_{km} = \begin{cases} \frac{2}{3} \times cluster\_radius \times No\_of\_service\_transformers, & load\_moment \leq 9643\ (LV) \\ 2 \times transformer\_radius \times No\_of\_service\_transformers, & load\_moment > 9643\ (MV) \end{cases} \tag{A18}$$

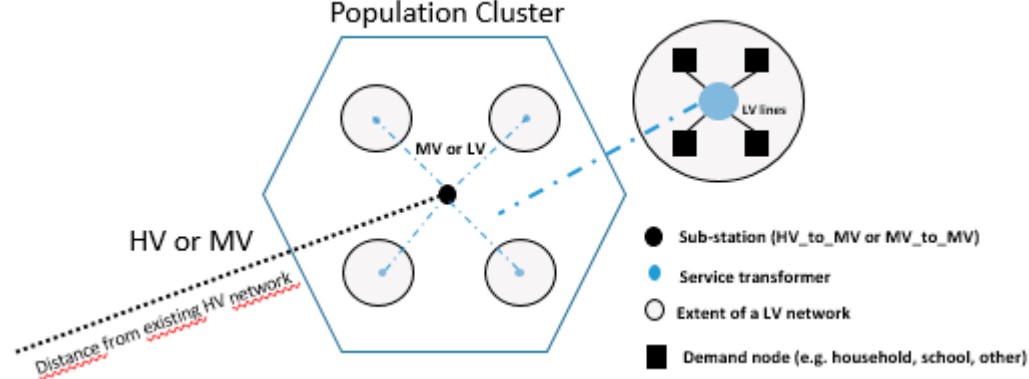

**Figure A4.** Estimating the size of transformers and their connection to sub-station.

Step 3. Sizing distribution lines (LV)

The area of each service transformer is then divided into a number of smaller circles (Figure A5) each one representing a demand node, assumed to be equally spaced within the larger circle. The distance between two demand nodes is defined as twice the radius of one of the smaller circles. The calculations do not consider the routing of LV lines from the transformer.

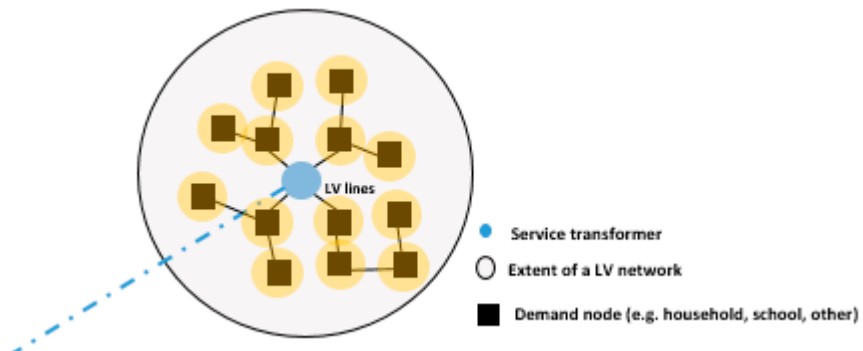

**Figure A5.** Sizing the LV network for each transformer in the population cluster.

The total length of LV lines per transformer is defined as described in (A19):

$$LV\_km\_per\_transformer = 2 \times r_{demand\ node} \times total\_nodes \quad (A19)$$

where:

$$r_{demand\ node} = \sqrt{\frac{demand\ node\ area}{\pi}} \quad (A20)$$

$$demand\ node\ area = \frac{transformer\_area\_coverage_{max}}{total\_nodes} \quad (A21)$$

Finally, the total number of distribution (LV) lines per cluster is estimated by (A22):

$$distribution\_line_{km} = LV\_km\_per\_transformer \times No\_of\_service\_transformers \quad (A22)$$

Step 4. Estimating the total investment cost for grid extension per cluster

In the last step, the total cost of grid extension per cluster is estimated by taking into account all partial costs as described in (A23):

$$
\begin{aligned}
grid\ &extension\ cost_{per\ cluster} \\
&= (transmission\_line_{km} \times transmision\_line\_cost) \\
&+ (connection_{linekm} \times connection\_line\_cost) \\
&+ (distribution_{linekm} \times distribution\_line\_cost \\
&+ (substation\_type \times substation\_cost) \\
&+ (No\_of\_service\_transformers \times service\_transformer\_cost) \\
&+ (total\_nodes \times node\_connection\_cost)
\end{aligned}
\quad (A23)
$$

**Appendix E. Detailed Results of Sensitivity Analysis**

The sensitivity analysis in this study was conducted in order to identify which are the most critical parameters and how they affect the least cost electrification mix and investment requirements. Six parameters were selected as shown in Table A5. Option 1 (or Baseline) includes the values as presented in previous paragraphs and used in the analysis so far. Option 2 includes modification of these values; for parameters 1–3 modifications intend to a more aggressive electrification strategy; for parameters 4–6 modifications suggest a cost increase in selected technologies. Finally, option 3 suggest an alternative approach to electricity demand targeted for each population cluster. The latter adopted an approach based on available poverty and GDP data (elaborate description in Appendix F). In total, ninety-six scenarios were generated and analysed.

**Table A5.** List of parameters used in the sensitivity analysis and their selected available options.

| # | Parameters | Option 1—Baseline | Option 2 | Option 3 |
|---|---|---|---|---|
| 1 | Population growth (PG) | 2.83% | 3.10% * | - |
| 2 | Electricity demand target (EDT) | Urban—Tier 4 Rural—Tier 1 | Urban—Tier 5 Rural—Tier 3 | Custom Residential Electricity Demand Indicative Target Layer (CREDIT) |
| 3 | Electrification rate in 2023 (ER23) | 50% | 80% | - |
| 4 | Grid generating cost of electricity (GGC) | 0.076 $/kWh | +25% | - |
| 5 | PV cost factor (PVC) | 0% | +25% | - |
| 6 | Diesel cost (DC) | 1.2 $/liter | 1.5 $/liter | - |

* Based on the highest variant of population growth as in [104].

Between all scenarios, the total investment requirements to achieve universal access to electricity in Malawi ranged between $1.65–7.78 billion. As seen in Figure A6, parameter 2 shows very low variance in all options studied. That is, parameter 2 is a quite strong determinant of electrification investment in comparison to the rest of parameters studied. Higher level of targeted electricity demand in population clusters rises significantly the total cost of electrification. Parameters 1, 3 and 6 do have a noticeable—yet not as strong—impact on the total investment; option 2 of these parameters indicates higher median value. For parameter 1 this is naturally explained by higher population growth, which also causes the min/max values to shift upwards. The second option for parameter 3 mandates the electrification of bigger part of population in the first five years; this results in higher penetration of off-grid systems which in turn are more capital intensive in terms of per unit capacity ($/kW). Higher diesel price leads to lower penetration of diesel based systems which are replaced either by other off-grid systems or grid connection; both alternatives have higher cost per capacity unit, explaining the variation observed in parameter 6. Finally, minor changes in total investment were observed by the variation of parameters 4 and 5.

The share of grid connected population ranges between 32.6–80.1%. Parameter 2 is the strongest determinant of grid penetration in the total mix, defining therefore the above limits. Parameters 3, 4 & 5 can induce a maximum of 1.3%, 1.2% and 3% increase in grid share respectively between options 1 and 2. No effect on grid share was observed by parameter 6. The share of stand-alone systems varies reversely with their share ranging between 10.2–64.7%. Mini-grids share ranges between 0–0.7% with the upper limit observed only when parameters 1, 2, 4 & 5 are set to option 2. The interplay between decentralized technologies is notably affected by parameter 5. Higher PV costs allow the penetration of other renewable off-grid technologies in the optimal mix; the cost of diesel affects the optimal mix only when parameter 5 is set at option 2, otherwise its impact is negligible.

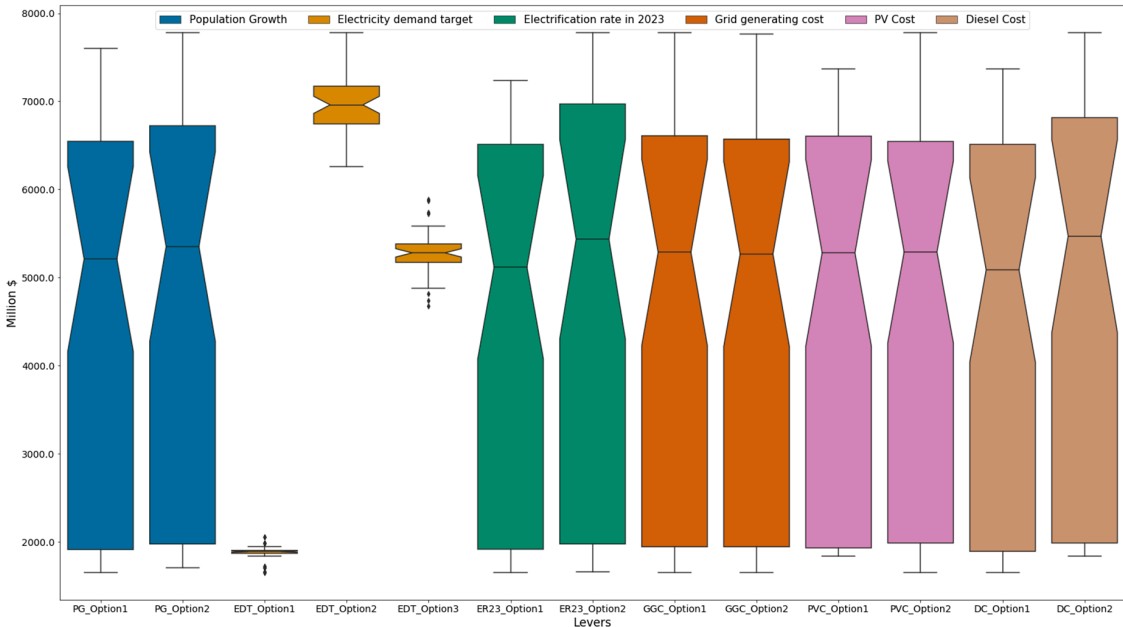

**Figure A6.** Investment variation for the achievement of universal electrification in Malawi as retrieved by the 96 scenarios developed in this study. The scenarios reflect the modification of six selected parameters and represent the impact of each one on the total investment.

## Appendix F. The Custom Residential Electricity Demand Indicative Target (CREDIT) Layer

A customized raster layer indicating residential electricity demand target over Malawi has been developed by using open access poverty and GDP maps as described in Section 2.1. First, an equal interval classification technique using five classes was applied on the poverty map; the breaking values indicated intervals between 0–100% of headcount poverty rate. The GDP map was classified based on geometric intervals since this technique is particularly useful for datasets that are not normally distributed; it creates a balance between highlighting changes in the middle values and the extreme values; therefore, a good fit for the GDP data available in this case. Then, the two layers were reclassified as shown in Table A6 and added under equal weighted factors (0.5) using raster calculation.

**Table A6.** Re-classification of GDP and poverty layers into five classes. $I_{1-5}$ are the geometric intervals of the classification process.

| Initial Poverty Layer | Poverty Classification | Initial GDP Layer | GDP Classification |
|---|---|---|---|
| $0 \leq poverty < 0.2$ | 5 | $0 < GDP < I_1$ | 1 |
| $0.2 \leq poverty < 0.4$ | 4 | $I_2 \leq GDP < I_3$ | 2 |
| $0.4 \leq poverty < 0.6$ | 3 | $I_3 \leq GDP < I_4$ | 3 |
| $0.6 \leq poverty < 0.8$ | 2 | $I_4 \leq GDP < I_5$ | 4 |
| $poverty \geq 0.8$ | 1 | $GDP \geq I_5$ | 5 |

The output provided an indicative demand target index ranging from 0 to 5; 0 indicating the lowest potential target and 5 the highest. Finally, using 1-D linear interpolation the above target index was translated into kWh/capita/year as shown in Figure A7. The interpolation was based on the multi-tier framework for energy access adapted to reflect the situation in Malawi; that is, the lowest and highest values were set at 8.8 and 680.2 kWh/capita/year for Malawi.

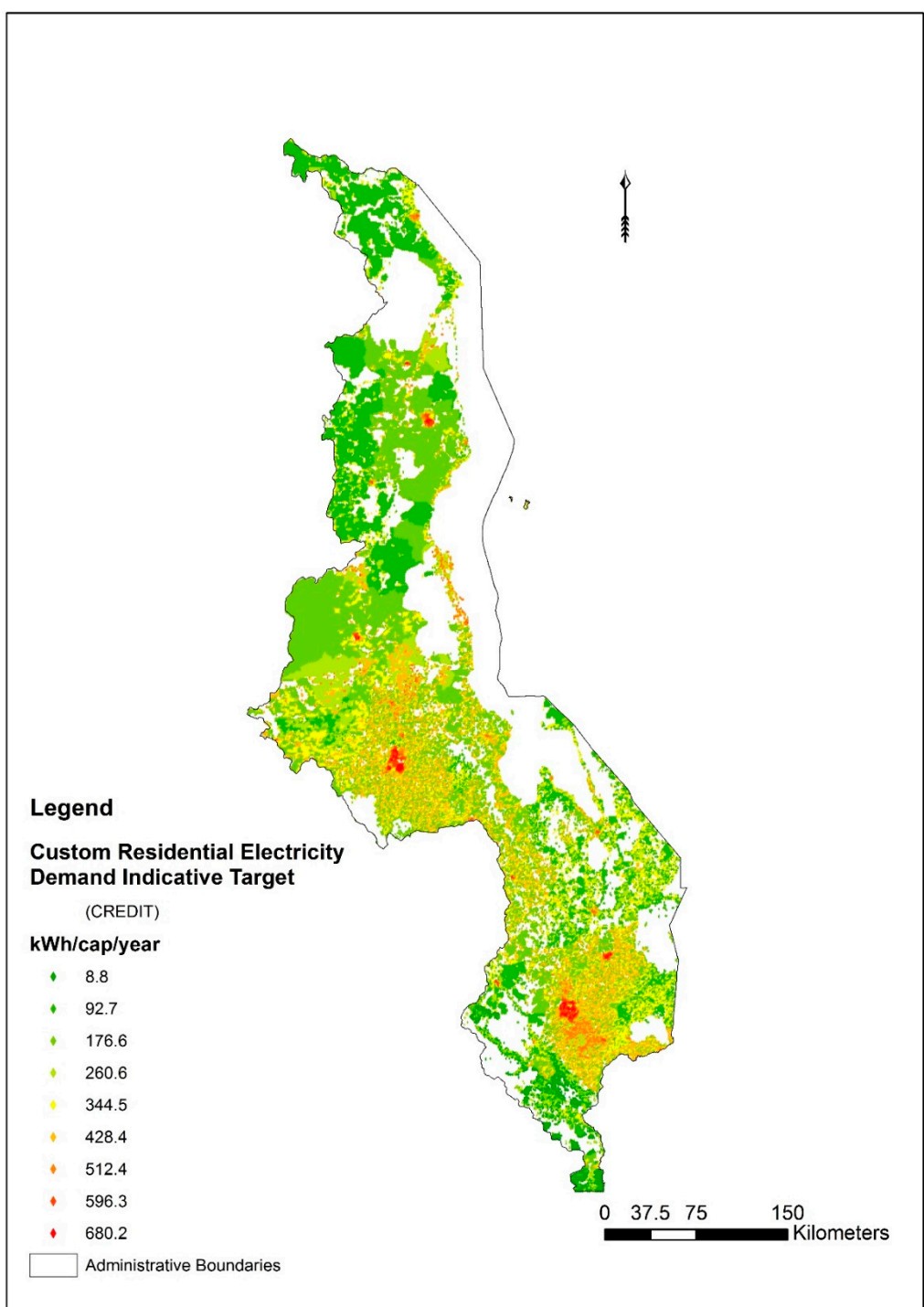

**Figure A7.** Customized layer indicating electricity demand target levels (in kWh/capita/year) over Malawi, based on openly available poverty and GDP maps.

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
