# Peer review of "The Role of Open Access Data in Geospatial Electrification Planning and the Achievement of SDG7. An OnSSET-Based Case Study for Malawi"

_energies, doi:10.3390/en12071395_

Round 1
Reviewer 1 Report
This paper is part of a research programme that the research team is undertaking and they have published quite a few papers using the same methodology. In terms of original contribution to knowledge, the paper does not contribute much except that a new case study (I've not seen Malawi case before) has been used. The other top-up material on GIS data and its usefulness is available in the literature and it does not make the work extra special.
I have not checked the paper for self-plagiarism but when same set of authors try to publish similar outputs in many platforms, this becomes a major issue. I suspect this can be an issue but I cannot confirm this without having the standard check results that the journal uses.
Data intensive models face the challenge of access to reliable data. This is an issue with many countries and even the official statistics cannot always be relied upon. In such a case, this sort of study needs to rely on alternative sources and the quality issue assumes importance here.
Otherwise, the paper is well-written and clearly presented. It meets the publication standards.
Author Response
Dear Reviewer,
Thank you for your comments/suggestions. A point to point response to your comments is available in the attached document. Glad to get your feedback and any additional modifications you feel can improve this paper.

Reviewer 2 Report
Review of the paper “The role of open access data in geospatial electrification planning and the achievement of SDG7. An OnSSET based case study for Malawi”
General comments
The paper is original and its content is very interesting. It aligns with the interests of the journal and I believe that it will attract the attention of readers in the field. The authors present and discuss a complete overview of open access geospatial data as well as GIS tools which constitute a necessity of any energy system model since it requires spatially implicit data to assess regional case studies. Moreover, the authors illustrate the capabilities of one particular open source tool “OnSESET” by applying it to the real case study of Malawi.
I recommend the article to be considered for publication, however, in my opinion the article contains a lot of information which makes it too long and difficult to follow. I think that it can be shortened so as to be more concise and better keep the readers’ attention. Minor comments and suggestions to be implemented as convenience of the authors below:
1) The paper is well written and well organised, however, I found it too long. I appreciate that the paper first presents the review of the open access data and GIS tools and then address a real case study. However, I think that the case study could be more concise, and I recommend shortening it. I would say that the main message to convey with the article is the role of open access and the article should focus on reviewing the existing data, tools and methods available to support the SDG7. In fact, in the current version, the title, introduction and discussion are focused on this main message.
2) I think that the paper would benefit a lot by including a framework figure summarising the steps to be followed when assessing any case study, from the gathering of data to the results analysis and mapping the main existing data, tools and methods. I suggest replacing the section 2.3 (questions) by a figure illustrating a conceptual framework for any electrification planning study using open access sources and tools.
3) In line with my comment 1) I suggest presenting the case study as an illustration of the applicability and capabilities of open data and GIS tools to guide decision and policy-makers towards ensuring the access to affordable, reliable, sustainable and modern energy for all.
a. I suggest shortening section 3.1 and 3.2 and move most of their content to an Annex while summarising the motivation to address the case study of Malawi.
b. I suggest moving the sensitivity analysis to one new Annex while summarising its outcome in the text by providing confident intervals on the deterministic results.
4) I suggest merging the discussion and conclusions sections and to sum up the main messages in a more concise way.

Author Response

(The authors gave the same response as above.)
